# Towards predicting dynamic stability of power grids with Graph Neural Networks

## Abstract

To mitigate climate change, the share of renewable energies in power production needs to be increased. Renewables introduce new challenges to power grids regarding the dynamic stability due to decentralization, reduced inertia and volatility in production. However, dynamic stability simulations are intractable and exceedingly expensive for large grids. Graph Neural Networks (GNNs) are a promising method to reduce the computational effort of analyzing dynamic stability of power grids. We provide new datasets of dynamic stability of synthetic power grids and find that GNNs are surprisingly effective at predicting highly non-linear targets from topological information only. We show that large GNNs trained on our rich dataset outperform GNNs from previous work, as well as several baseline models based on handcrafted features. Furthermore, we use GNNs to demonstrate the accurate identification of particularly vulnerable nodes in power grids, so called *troublemakers*. Lastly, we find that GNNs trained on small grids generate accurate predictions for on a large synthetic model of the Texan power grid, which illustrates the potential real-world applications of the presented approach.

## 1 Introduction

Adaption to and mitigation of climate change jointly influence the future of power grids: 1) Mitigation of climate change requires power grids to be carbon-neutral, with the bulk of power supplied by solar and wind generators. These are more decentralized and as opposed to conventional turbine generators they have no intrinsic ability to respond to power imbalances and frequency deviations. Furthermore, the production of renewables is more volatile. Renewable energies will have to start contributing to the dynamical stability of the system (Milano et al., 2018; Christensen et al., 2020) in the future, requiring a new understanding of the complex synchronisation dynamics of power grids. 2) A higher global mean temperature increases the likelihood as well as the intensity of extreme weather events such as hurricanes or heatwaves (Field et al., 2012; Pörtner et al., 2022) which result in great challenges to power grids. Building sustainable grids as well as increasing the resilience of existing power grids towards novel threats are challenging tasks on their own. Tackling climate change in the power grid sector calls for a solution to both at the same time and requires new methods to investigate aspects of dynamic stability.

Power grids are complex networks, consisting of nodes that represent different producers and consumers, as well as edges that represent power lines and power transformers. In contrast to many other networks, the interaction of nodes through the edges is governed by physical equations, the power flow. Their emergent properties can be highly unintuitive. For example, the Braess paradox describes the phenomenon that adding lines to a power grid may reduce its stability ((Witthaut & Timme, 2012; Schäfer et al., 2022)). Such effects can be non-local, i.e. the parts of the grid with decreased stability might be far away from the added line. Similarly, failures of a line in one part of the network can lead to overloads far away. Our work deals with the challenge of predicting the ability of the grid to dynamically recover after localized faults perturbed the system.

Classically, the dynamical actors are connected to the highest voltage level, the transmission grid. Transmission grid operators routinely simulate potential faults in the current state of the power grid, to assess its real time dynamic resilience. The possible faults are called contingencies in this context. Even though such simulations do not explicitly model the lower voltage layers of the grid, they are already compute bound. Hence, not all contingencies of interest can be tested for. As distributed re-

newable generation is typically connected at lower grid levels, this problem will become more acute as renewables start playing a larger role in the grids dynamics. Conducting high-fidelity simulations of the whole hierarchy of the power grid and exploring all states will not be feasible (Liemann et al., 2021). For future power grids, knowledge of how to design robust dynamics is required. This has led to a renewed interdisciplinary interest in understanding the collective dynamics of power grids (Brummitt et al., 2013), with a particular focus on the robustness of the self-organized synchronization mechanism underpinning the stable power flow (Rohden et al., 2012; Motter et al., 2013; Dörfler et al., 2013) by physicists and control mathematicians Witthaut et al. (2022).

Synchronization refers to the fact that a stable power flow requires all generators to establish a joint frequency. It is self-organized in the sense that this is achieved without further communication or an external signal. To understand which structural features impact the self-organized synchronization mechanism, it has proven fruitful to take a probabilistic view (Menck et al., 2014; Hellmann et al., 2016). Probabilistic approaches are well established in the context of static power flow analysis (Borkowska, 1974). In the dynamic context, considering the probability of systemic failure following a random fault effectively averages over the various contingencies. Such probabilities are thus well suited to reveal structural features that enhance the system robustness or vulnerability. This approach has been highly successful in identifying particularly vulnerable grid regions (Menck et al., 2014; Schultz et al., 2014a; Nitzbon et al., 2017) and revealing general mechanisms of desynchronization (Hellmann et al., 2020). Probabilistic stability assessments recently gained more attention in the engineering community as well (Liu & Zhang, 2017; Liu et al., 2019; Liemann et al., 2021).

Given the need for probabilistic analysis, and the computational cost of explicit simulations, we apply Graph Neural Networks (GNNs) to directly predict probabilistic measures from the system structure. Such GNNs could be used to select critical configurations for which a more detailed assessment should be carried out. Moreover, the analysis of the decision process of ML-models might lead to new unknown relations between dynamical properties and the topology of grids. Such insights may ultimately inform the design and development of power grids. Since datasets of probabilistic stability in power grids of sufficient size do not exist yet, we introduce new datasets, that consist of synthetic models of power grids and statistical results of dynamical simulations. We simulated datasets of increasing complexity to get closer to reality step by step. There are 10,000 small grids, 10,000 medium-sized grids and for evaluation purposes one large grid based on a synthetic Texan power grid model.

**Related work on power grid property prediction** Since power grids have an underlying graph structure, the recent development of graph representation learning (Bronstein et al., 2021; Hamilton, 2020) introduces promising methods to use machine learning in the context of power grids. There is a number of applications using GNNs for different power flow-related tasks (Donon et al., 2019; Kim et al., 2019; Bolz et al., 2019; Retiére et al., 2020; Wang et al., 2020; Owerko et al., 2020; Gama et al., 2020; Misyris et al., 2020; Liu et al., 2021; Bush et al., 2021; Liu et al., 2020; Jhun et al., 2022) and to predict transient dynamics in microgrids (Yu et al., 2022). In (Nauck et al., 2022) small GNNs are used to predict the dynamic stability on small datasets. The authors demonstrate the general feasibility of the approach, but do not compare to conventional baselines. We add such baselines and introduce datasets that have ten times as many grids. This allows us to train GNNs with much higher capacity to achieve higher predictive power.

**Our main contributions are:** We introduce new datasets of probabilistic dynamic stability of synthetic power grids. The new datasets have 10 times the size of previously published ones and include a Texan power grid model to map the path towards real-world applications. We also observe a relevant new class of nodes in the dataset: So-called *troublemakers*, at which perturbations are strongly amplified. Such nodes may be dangerous to hardware and the overall grid stability. Their identification constitutes an additional task. We train strong baselines and benchmark models to evaluate the difficulty of all tasks. Our results demonstrate i) that the larger dataset allows to train more powerful GNNs, (ii) which outperform the baselines, and (iii) transfer from the new datasets to a real-sized power grid. The general approach is visualized in Figure 1.

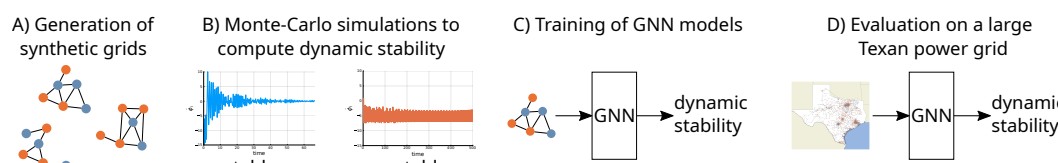

A) Generation of synthetic grids  B) Monte-Carlo simulations to compute dynamic stability  C) Training of GNN models  D) Evaluation on a large Texan power grid

stable    unstable

GNN → dynamic stability

GNN → dynamic stability

Figure 1: We generate new datasets of the dynamic stability of power grids, based on synthetic power grids (A) and statistics of dynamical simulations (B). Then, GNN models are trained to predict the dynamic stability of the synthetic grids (C) and evaluated on a Texan power grid model (D).

## 2 GENERATION OF THE DATASETS

**Structure and modeling of power grids**   Viewed as networks power grids have distinctive topological properties. Power grids are sparsely connected. The degree distribution has a local maximum at very small degrees (e.g. $\approx 2.3$), an exponentially decaying tail and a mean of $\approx 2.8$ (Schultz et al., 2014b). Hence, most nodes are only connected to a small number of neighbours though a few highly connected nodes typically do exist.

The study of non-linear self-organizing processes in power grids, specifically the interplay of synchrony and active power flows has been an active topic in the research of complex power grids (Witthaut et al., 2022). Synchrony is crucial for power grids as a stable power flow is only possible if all generators operate at the same frequency. At the same time the ever present fluctuations in demand have to be compensated by locally varying the frequency. The interplay between energy balance and frequency throughout the grid is self-organized as it happens without further communication between the nodes or some central entity. The relevant timescale of this process is seconds, and we will analyze the capability of the grid to regain synchrony following a localized fault.

**Power grids as dynamical systems**   A full scale analysis that can treat high-fidelity models of real systems is currently out of reach for several reasons. These include that real world data does not exist or is not accessible, synthetically generating large numbers of realistic grids is challenging, and that large dynamical models can not be simulated fast enough with current software (Liemann et al., 2021). These problems force trade-offs on us, most notably reducing the details of the intrinsic behaviors of dynamical actors to that of inertial oscillators. This leads to the Kuramoto model Kuramoto (1975), a paradigmatic model for synchronization studies (Kuramoto, 2005; Rodrigues et al., 2016), which was introduced for power grids in (Bergen & Hill, 1981). This model strikes a careful balance of capturing the key dynamics that govern synchronization in real power grids while remaining computationally - and to some degree analytically - tractable. It is thus highly useful for understanding the relationship between grid structure and synchronization, but it should not be taken to be a fully adequate model of the grid by itself. Any future treatment of dynamic stability based on more accurate models will also have to solve the challenging sub-problem of the impact of topology on synchrony that we consider here, thus we see this work as a first step.

In an ideal AC grid, the nodes voltage is given by a 50 or 60Hz sine curve. Writing $\phi_i$ for the deviation of the phase of the local AC voltage from an arbitrary reference signal, the instantaneous power flow on line $i, j$ ($P_{ij}$) is given in terms of the line parameter $K$ by $P_{ij} = K \sin(\phi_i - \phi_j)$. The normalized time derivative $\frac{1}{2\pi}\dot{\phi}_i$ provides the local frequency deviation from the reference frequency. The two most important dynamical processes that establish synchrony and a stable power flow are I) inertia, i.e. the change of local frequency as a result of absorbing a local power imbalance $M\ddot{\phi}_i = \Delta P_i$, and II) droop control, the local change of injected power due to frequency deviation $P_i^{\text{droop}} = -\alpha\dot{\phi}_i$.

In order to extract the impact of topology as cleanly as possible we assume homogeneous inertia constant $M$, droop parameter $\alpha$ and line parameters $K$. The dynamical equations for the self-organized synchronization and stabilization of the active power flow are then given by conservation of energy. Let $P_i^d$ denote the power injected/consumed at node $i$ and $A_{ij}$ the adjacency matrix:

$$\Delta P_i = P_i^d + P_i^{\text{droop}} - \sum_j A_{ij} P_{ij} \tag{1}$$

$$M\ddot{\phi}_1 = P_i^d - \alpha\dot{\phi}_i - K \sum_j A_{ij} sin(\phi_i - \phi_j) \tag{2}$$

Synchronous operation requires $0 = \ddot{\phi}_i = \dot{\phi}_i$. The fixed point equation $P_i^d = K \sum_j A_{ij} sin(\phi_i^* - \phi_j^*)$ is the power flow equation on a grid with topology given by the adjacency matrix $A$.

**Quantifying dynamic stability of power grids**    We quantify dynamic stability with the single-node basin stability (SNBS) (Menck et al., 2014). This measure is widely used in the study of synchronization phenomena. As mentioned in the introduction, it is probabilistic, i.e. defined as the probability that the system recovers following a perturbation by a random fault.

In the dynamical systems community this perturbation is typically modeled by a distribution of initial conditions of the post fault system. For every node $i$ we choose a distribution of initial conditions $\rho_i$ corresponding to contingencies localized at that node. Then the SNBS at node $i$ is the probability that the entire system returns to the fixed point from an initial condition drawn from $\rho_i$. Call $p(y|x)$ the probability of $y$ given the initial condition $x$. For a distribution of initial conditions $\rho(x)$ we write $p(y|x \sim \rho) = \int p(y|x)\rho(x)$ for the compound distribution. Then we have:

$$\text{SNBS}_i = p\left(\phi(\infty) = \phi^* \mid (\phi(0), \dot{\phi}(0)) \sim \rho_i\right) \tag{3}$$

This probability can be estimated as the outcome of a Bernoulli experiment. Here we simulate 10,000 trajectories per node to minimize statistical errors. The underlying simulations to generate the datasets are feasible for anyone with some domain knowledge, but the composition of entire datasets require significant amounts of computational resources. For the datasets with 20,000 grids and the Texan power grid, the simulations take roughly 700,000 CPU hours. Thus an important contribution is to publish a full dataset to enable groups that have less computational resources to work on this important problem. The goal of our work is the usage of GNNs to replace these expensive simulations. To that end, we train GNNs to learn the graph function $(P, A) \rightarrow \text{SNBS}$ where $P_i$ is a featurized version of $P_i^d$

## 2.1 New troublemaker definition

The condition in the SNBS is not sufficient to ensure stable grid operation at all times. If the transient trajectory after the perturbation violates operational bounds, machines in the system switch off to protect themselves, potentially triggering failure cascades and large blackouts. This motivates the definition of survivability (Hellmann et al., 2016), the probability that the system stays within these bounds after a fault. In our dataset, we observed that there are some nodes for which the transient reaction is vastly larger than the initial perturbation. We define the maximum frequency deviation (MFD) of the whole system from an initial condition as

$$\text{MFD}_i\left(\phi(0), \dot{\phi}(0)\right) = \max_{t, j}\left|\dot{\phi}_j(t)\right|. \tag{4}$$

We consider nodes as safe if they have a very small probability $\gamma$ that the MFD is larger than a critical threshold $\beta$ following a fault, other nodes are troublemakers (TM)

$$\text{TM}_i = \begin{cases} 0 & \text{if } p\left(\text{MFD}_i < \beta \mid (\phi(0), \dot{\phi}(0)) \sim \rho_i\right) > 1 - \gamma \\ 1 & \text{otherwise.} \end{cases} \tag{5}$$

The minimum size that can be chosen for the rare event rate $\gamma$ is determined by the statistical error of the estimator of the probability, see Appendix A.5.

For choosing the critical threshold $\beta$ consider, that the outer limits for frequency deviations for the European grid are $+2$Hz or $-3$Hz. For our model we chose the symmetric critical threshold $\sim 2.4$Hz, corresponding to a maximum deviation of the angular velocity of $|\dot{\phi}| < 15 =: \beta$, both for simplicity and comparability to previous work. We tune the distribution of initial conditions to focus on the worst offenders, settling on $\rho_i$ confined to $\pm 0.4$Hz, corresponding to an amplification by a factor of $\frac{2.4Hz}{0.4Hz} = 6$. More details are given in Appendix A.5. The observation of large amplifications and the target function are both of practical importance and novel as far as we are aware.

## 2.2 Modeling of the Texan power grid

To take a further step towards real-world applications, we evaluate the performance of our GNN models by analyzing the dynamic stability of a real-sized synthetic power grid. Real power grid data are not available due to security reasons and calculating an entire SNBS assessment of the fully parameterized synthetic model by Birchfield appears not to be feasible due to the computational effort (Liemann et al., 2021). We chose a synthetic model derived from the Texan power grid topology, introduced by Birchfield et al. (2017). The synthetic Texan power grid model consists of 1,910 nodes after removing 90 nodes that are not relevant for the dispatching.

We use the same modelling approach as for the other grids, i.e we use only the topological properties. As a consequence, we only investigate the potential applicability of GNNs to real-sized grids and can not make any statements about the real-world Texan power grid. Even after applying the simplifications, the simulations are already very expensive due to the large number of nodes. To manage the computational cost of simulating dynamic stability of such a large grid, we reduce the number of simulated perturbations from 10,000 to 1,000. Nonetheless, the simulation of that grid takes 127,000 CPU hours. Computing less perturbations results in an increased standard error of approximately $\pm 0.031$ for the SNBS estimates.

## 2.3 Structure of the datasets

To generate the datasets, we closely follow the methods in Nauck et al. (2022) and extend their work by computing 10 times as many grids. To investigate different topological properties of differently sized grids, we generate two datasets with either 20 or 100 nodes per grid, referred to as dataset20 and dataset100. To enable the training of complex models, both datasets consist of 10,000 graphs. Additionally, one large synthetic Texan grid is provided for testing out-of-sample performance, see Section 2.2.

For every grid two input features are given, namely the adjacency matrix $A \in \{0, 1\}^{N \times N}$ representing the topology and a binary feature vector $P_i^d \in \{-1, 1\}^N$, specifying the power injection/demand at the nodes, cf. Equation (1). Here $N$ is the number of nodes. Likewise, for every grid two target vectors are given, SNBS $\in [0, 1]^N$, see Section 2, and the TM $\in \{0, 1\}^N$. The TM class is derived from the maximum frequency deviation MFD $\in [0, \infty]^N$, which is also available in the dataset, see Section 2.1. Details regarding the modeling strategy are provided in Appendix A.4. Examples of the grids of dataset20, dataset100 and the Texan power grid as well as the distributions of SNBS (characterized by multiple modes) and the troublemaker nodes TM (imbalanced binary classification task derived from the MFD) are given in Figure 2. Even though the same modelling approach is used, there are significant variations that are entirely caused by different topologies and grid sizes. Interestingly, the SNBS distribution of Texan power grid has a third mode which is challenging for prediction tasks. Overall, the power grid datasets consist of the adjacency matrix encoding the topology, the binary injected power $P$ per node as input features, and nodal SNBS and MFD.

## 3 Predicting dynamic stability of power grids using GNNs

In this section, we predict the dynamic stability on the new datasets using GNNs. We start by introducing the experimental setup, followed by the prediction of SNBS on the two grid sizes as well as analyzing out-of-distribution capabilities from small to large grids. Subsequently, we establish the advantages of our large dataset in comparison to previous work. Afterwards, we evaluate the GNNs trained on our datasets on the Texan power grid as a larger, more realistic test-case. Lastly, we identify troublemakers.

### 3.1 Experimental setup

We train GNNs on a nodal prediction tasks, using regression for SNBS, and regression with thresholding as well as classification for TM (visualized in Figure 3). As input the GNNs are given an adjacency matrix and the power injection/demand at the nodes, cf. Section 2.3. We split the datasets in training, validation and testing sets (70:15:15). The validation set is used for the hyperparameter optimization, we report the performance on the test set. To minimize the effect of initializations we use 5 different initializations per model and compute average performances using the three best.

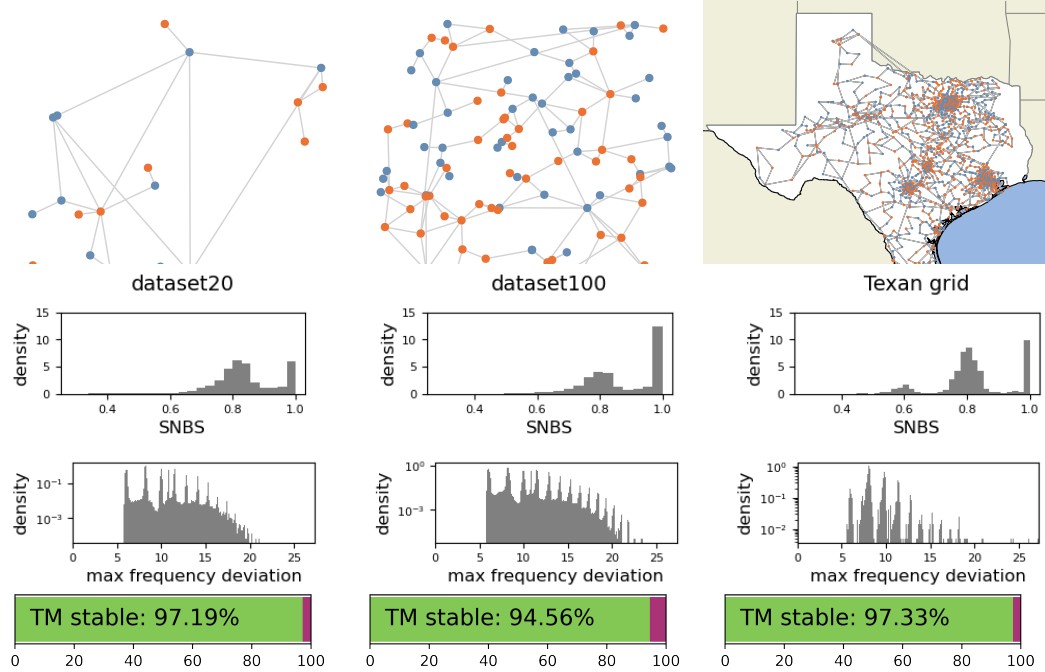

Figure 2: Examples of power grids in the datasets with 20 nodes (top left) and 100 nodes (top center) and the Texan power grid model (top right). Blue color denotes sources and the orange sinks. Below, the histograms of SNBS and the maximum frequency deviations (logarithmic scale) are shown. At the bottom, the share of troublemakers (TM) is shown, where green represent stable nodes and purple troublemakers (TM).

To evaluate the robustness of the GNNs, we analyze the performance of different models based on several GNN-architectures: GNNs with ARMA filters by Bianchi et al. (2021), Graph Convolutional Networks (GCN) by Kipf & Welling (2017), SAmple and aggreGatE (SAGE) by Hamilton et al. (2017) and Topology Adaptive Graph Convolution (TAG) by Du et al. (2017). We refer to the models by ArmaNet, GCNNet, SAGENet and TAGNet. We conduct hyperparameter studies to optimize the model structure regarding number of layers, number of channels and layer-specific parameters using dataset20. Afterwards, we optimize learning rate, batch size, scheduler for dataset20 and dataset100 separately. Details on the hyperparameter study and the models are given in Appendix A.7.

**Baseline models** To better assess the GNN performance, we set up several baseline models. Schultz et al. (2014a) were the first to attempt predicting nodal dynamic stability of power grids using a logistic regression of common network measures and hand-crafted features. For the first baselines, we set up similar linear regression models with the following input features: degree, average-neigbor-degree, clustering-coefficient, current-flow-betweenness-centrality, closeness-centrality and the injected power $P^d$. Additionally, we use more complex Multilayer perceptrons (MLPs) trained on the same features as baselines. We investigate the performance of two differently sized MLPs. MLP1 has one hidden layer with 35 units per hidden layer, resulting in 1,541 parameters and MLP2 consists of 6 hidden layers and 500 hidden units per layer leading to 1,507,001 parameters. We conducted hyperparameter studies to optimize the batch sizes and learning rates.

**Metrics for evaluation** To analyze the performance, we use the coefficient of determination ($R^2$-score) for regression and F2-score for classification. F2-score is a modified F-score giving more weight to recall and less to the precision in the calculation of the score. In case of identifying vulnerabilities of power grids it is more important to identify all critical states, even if this increases the number of false positives. The details of computing the metrics are provided in Appendix A.6.

### 3.2 PREDICTION OF SNBS

**GNNs can accurately predict SNBS** In our first set of benchmark experiments, the goal is to predict SNBS with high accuracy, as measured by the coefficient of determination $R^2$. The key

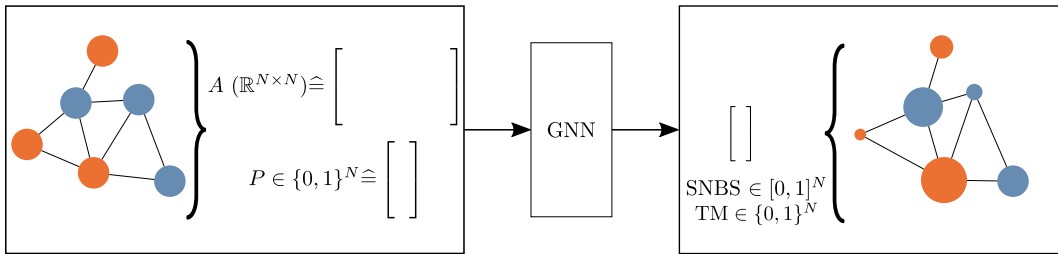

Figure 3: Prediction of nodal outputs SNBS and TM using GNNs. The inputs are the adjacency matrix $A$ and the injected power $P^d$. The prediction is purely based on the structure and topology of the grid and does not consider individual faults.

result is the surprisingly high performance of GNNs across all datasets, see the first two columns in Table 1. $R^2$ reaches values above 82 % for dataset20 and above 88 % for dataset100. SNBS is a highly nonlinear property and the obtained performance exceeds expectations. The predictive performance captures not only the general trend but the modalities in the data as well (cf. Figure 4). Interestingly the previously published Ar-bench (Nauck et al., 2022) performs worse than the MLPs, but the larger GNNs outperform all baselines.

Table 1: Results of predicting SNBS represented by $R^2$ score in %. Each column represents the evaluation on a different test set, e.g. *tr20ev20* denotes that the models are trained and evaluated using dataset20. Additionally, we analyze the out-of-distribution capabilities by evaluating the models on different datasets without retraining, e.g. we train a model on dataset20 and evaluate it on dataset100 and refer to this by *tr20ev100*. Besides the performance of the GNNs, we show the performance of a linear regression and Multilayer perceptrons using hand-crafted features as baselines.

| model | tr20ev20 | tr100ev100 | tr20ev100 | tr20evTexas | tr100evTexas |
|---|---|---|---|---|---|
| Ar-bench | $51.82 \pm 2.388$ | $60.34 \pm 0.299$ | $38.80 \pm 1.327$ | $38.55 \pm 5.897$ | $55.22 \pm 6.349$ |
| ArmaNet | $80.63 \pm 0.848$ | $87.47 \pm 0.073$ | $\mathbf{66.75} \pm 1.500$ | $60.54 \pm 5.622$ | $73.47 \pm 2.786$ |
| GCNNet | $70.64 \pm 0.262$ | $75.49 \pm 0.276$ | $59.46 \pm 0.450$ | $2.27 \pm 6.210$ | $-46.02 \pm 1.575$ |
| SAGENet | $65.46 \pm 0.208$ | $75.57 \pm 0.228$ | $52.27 \pm 0.784$ | $33.47 \pm 0.509$ | $55.38 \pm 1.390$ |
| TAGNet | $\mathbf{82.49} \pm 0.455$ | $\mathbf{88.22} \pm 0.135$ | $66.10 \pm 0.508$ | $\mathbf{62.69} \pm 2.029$ | $\mathbf{84.41} \pm 0.947$ |
| linreg | 41.75 | 36.29 | 5.98 | -11.39 | -22.62 |
| MLP1 | $58.47 \pm 0.149$ | $63.59 \pm 0.051$ | $28.49 \pm 1.493$ | $-34.52 \pm 17.934$ | $19.79 \pm 8.659$ |
| MLP2 | $58.20 \pm 0.0422$ | $65.52 \pm 0.038$ | $19.65 \pm 2.109$ | $5.81 \pm 10.58$ | $58.46 \pm 0.480$ |

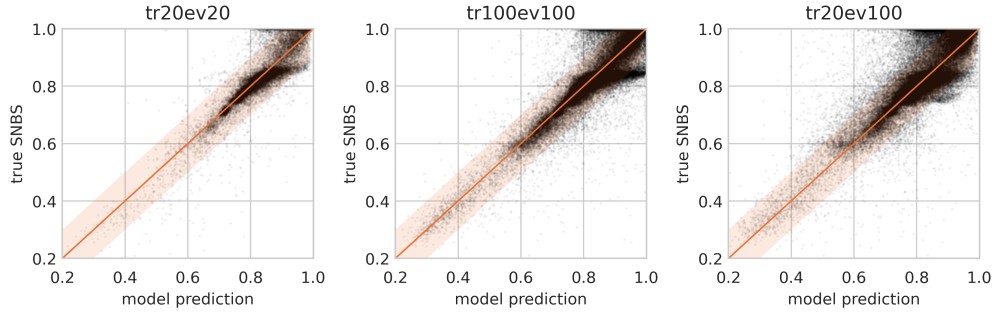

Figure 4: SNBS over predicted output of the ArmaNet model for tr20ev20 (trained on dataset20 and evaluated on dataset20), tr100ev100 and tr20ev100. The diagonal represents a perfect model ($R^2 = 1$), the banded region indicates predictions which are accurate up to an error of $\pm 0.1$.

**GNNs generalize from dataset20 to dataset100** Note that, as reviewed in the introduction, the dynamical properties of the power grid are non-linear and non-local. Our perturbations are localized, but the probabilistic measures look at the whole system response. Thus it is a priori unclear how well we can expect the models to generalize from smaller grids to larger ones. Training on small grids without loss of generalization and predictive power would be a huge advantage to scale to real power grids. To evaluate the potential of our datasets and GNN models to that end, we apply an

out-of distribution task by training the models on dataset20 and evaluating the performance without any further training on dataset100. The third column in Table 1 (tr20ev100) show that all GNNs generalize well and are able to predict SNBS with $R^2$ exceeding 66 %. We would like to emphasize the significance of that finding. Given sufficient size and complexity in the source dataset, GNNs can robustly predict highly nonlinear stability metrics for grids several times larger than the source. We did not expect grids of size 20 to be large enough to contain enough relevant structures to generalize to larger grids. Generalizing from small, numerically solvable grids to large grids is key for real world application. The computational cost of the dynamic simulations grows faster than linearly with the size of the grids, so computational time can be saved when training models on smaller networks or sections of real-sized grids. In comparison to the baselines, the generalization capabilities of the new GNN models are much better.

**Training on more data increases the performance of all models**   General Machine Learning convention assumes that larger dataset size allows to train larger models to higher performance. In this section, we investigate the influence of the size of the training set to show the relevance of the larger datasets. We train the models on the smaller dataset introduced in Nauck et al. (2022) after specifically optimizing the learning rate for the analysis of the smaller training set. Our experiments show that training on less data results in lower performance, see Figure 5. Instead of peak values of $R^2$ of 82.49 %, we only obtain 74.77 % for dataset20 and only 83.92 % instead of 88.22 % for dataset100. The results of all models are given in Appendix A.9. Comparing the performance differences on dataset20 and dataset100, the improvements are larger for dataset20. A reasonable explanation is the total number of nodes used for the training.

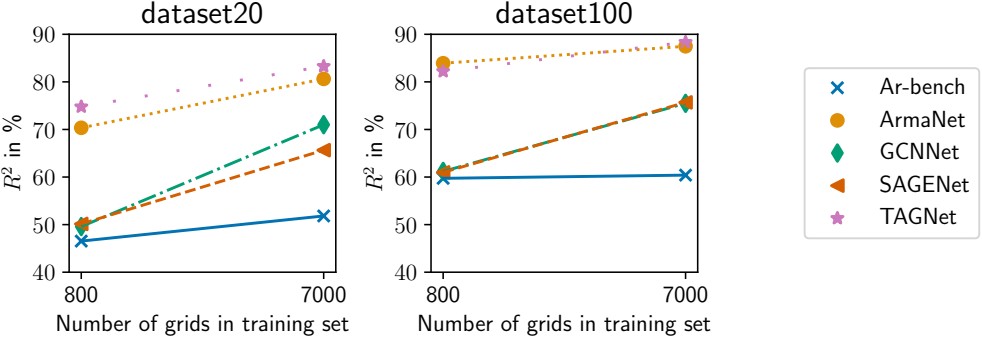

Figure 5: Comparison of the performance based on the size of the training set using 800 or 7,000 grids. Training on our larger dataset improves performance on all models. The 800 grids used for the training follow Nauck et al. (2022) and all models are evaluated on the newly introduced test set.

**Predicting SNBS on a Texan power grid model using the previously trained models**   Using GNNs for SNBS prediction becomes feasible, if they can be trained on relatively simple datasets and still perform well on large, complex grids. As an example of a large and complex grid, we use a Texan power grid model and evaluate the models previously trained on dataset20 and dataset100. The benchmark models achieve surprisingly high performance with $R^2$ values (above 84 %) and GCN as a sole exception, see the colums tr20evTexas and tr100evTexas in Table 1. Details on the poor performance of the GCN is given in Appendix A.10. Hence, the approach of training models on grids which are smaller by more than one order of magnitude is feasible. We want to emphasize that one successful attempt of a real-sized power grid should illustrate the general potential of this approach, but we still consider the hard evidence to be the generalization from 20 to 100. The performance is significantly better for the models trained on dataset100. We hypothesize that the repetition of geometrical structures more prevalent in dataset100 is useful for even larger grids. Grids of size 20 might still be to small to generalize to large grids, but the size 100 might actually be sufficient for many applications.

## 3.3   IDENTIFICATION OF TROUBLEMAKERS

In this section we introduce a further benchmark tasks, namely to classify nodes into two categories, stable nodes or troublemakers as defined in Section 2.1. As noted in Equation (5), this target is essentially a thresholded version of the maximum frequency deviation (MFD). Thus, we can either

directly train the classification task, or we can regress the MFD estimator and then threshold. Both strategies work and depending on the model different approaches seem to be best, see Table 2. The overall performance is very high, so the prediction of TM is feasible. In the main section, we only show the F2-score and the better performing baselines. Appendix A.11 contains the full results including different performance indicators. The GNNs ArmaNet and TAGNet outperform the baselines and especially TAGNet achieves a good performance.

Table 2: F2-score in % for TM prediction. The column *type* shows if classification (C) or regression (R) is used for training. For regression, thresholding at 1 is applied to compute the F2-score. For MLP2, training on dataset100 was successful with only one of the five seeds. We report that result as over-estimation of a strong baseline.

| model | type | tr20ev20 | tr100ev100 | tr20ev100 | tr20evTexas | tr100evTexas |
|---|---|---|---|---|---|---|
| ArmaNet | C | $86.85 \pm 0.503$ | $95.75 \pm 0.049$ | $84.94 \pm 4.050$ | $82.43 \pm 1.747$ | $85.14 \pm 1.335$ |
| TAGNet | C | $\mathbf{89.78} \pm 0.079$ | $\mathbf{96.36} \pm 0.063$ | $88.77 \pm 0.309$ | $82.78 \pm 2.122$ | $92.00 \pm 1.585$ |
| ArmaNet | R | $81.68 \pm 0.509$ | $88.29 \pm 5.898$ | $\mathbf{91.49} \pm 0.740$ | $87.15 \pm 2.922$ | $91.86 \pm 1.058$ |
| TAGNet | R | $84.55 \pm 1.136$ | $95.13 \pm 0.316$ | $90.20 \pm 1.624$ | $\mathbf{90.18} \pm 1.645$ | $\mathbf{95.48} \pm 0.562$ |
| linreg | R | 72.69 | 91.51 | 91.32 | 73.22 | 93.75 |
| MLP1 | R | $74.33 \pm 0.074$ | $91.60 \pm 0.006$ | $81.60 \pm 2.155$ | $75.17 \pm 2.837$ | $41.96 \pm 16.52$ |
| MLP2 | R | $74.38 \pm 0.036$ | 91.61 | $77.46 \pm 7.020$ | $68.89 \pm 5.248$ | 93.75 |

### 3.4 BENEFITS OF USING GNNs TO PREDICT DYNAMIC STABILITY

As the experiments above show, GNNs are suitable for the analysis of the dynamic stability, both in terms of computational effort and predictive power. The performance exceeded our expectations. For SNBS and the best model (TAGNet) roughly 95 % of the estimations differ only by 0.1 from the target values for tr100ev100 and tr100evTexas. This corresponds to the banded region shown in Figure 4. The threshold of 0.1 can be motivated by considering the distributions in Figure 2, because the modes are separated by 0.1. To achieve a similar accuracy by conducting dynamical simulations with Monte-Carlo sampling, 100 perturbations per node are needed, which is a significant reduction to 10,000 perturbations we used for the generation of the datasets. With 100 perturbations, the computation for one grid of size 100 takes roughly 30 minutes and for the Texan grid 530 days using one CPU. Evaluating the GNNs takes less than one second per grid. Hence using GNNs is more than 1,800 times faster for grids of size 100 and $4,6 \times 10^7$ times faster for the synthetic Texan power grid. This demonstrates the potential of analyzing a large number of different configurations using GNNs. Furthermore, we show the successful prediction of TM, which can motivate future applications.

## 4 CONCLUSION AND OUTLOOK

In this work, we analyze the probabilistic dynamic stability of synthetic power grids using GNNs. We generate new datasets that are 10 times larger than previous ones to enable the training of high capacity GNNs. Our benchmark results significantly improve over previous work and show that highly nonlinear SNBS can be predicted at surprisingly high accuracy by using more complex models and training on our larger dataset. We show that the models trained on our datasets can be used for prediction on a real-world-sized Texan power grid model. The results indicate the potential benefits of machine learning in this important domain. We expect the datasets to attract attention by research groups working on complexity science and non-linear dynamics as well. Furthermore, we introduce a new method to identify troublemakers in power grids and and show their successful prediction. Besides further improving the performance, future work might try to explain the decision process of GNNs to generate new insights on the relation of topology and stability. Encouraged by our results, we will continue to extend our datasets with increasingly more complex and realistic grids, aiming at real power grids. All data and code will be published upon publication (see Appendix A.1). Given sufficient computational resources, the code can easily be adapted to generate more training data, or to simulate grids of different sizes. For reviewing, the SNBS training is provided in a zip container at: `https://drive.google.com/drive/f olders/19hHqgup12xypYnS0ceEDaZRhqdqUG7gi?usp=sharing`. The open access enables the community to develop new methods to analyze future renewable power grids.

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

# A APPENDIX

This section includes additional information to generate the datasets, reproduce the presented results and additional results that are not already shown in the main section.

We start by providing information on the availability of the datasets and the used software, before providing more details regarding the modeling for the dataset generations. Subsequently, details regarding the training, the evaluation as well as the hyperparameter study are provided. Afterwards additional results are shown.

## A.1 AVAILABILITY OF THE DATASETS

The new datasets and full code for the training, evaluation and generation of the figures will be published upon publication on Zenodo and GitHub and it will be licensed under CC-BY 4.0 to enable the community to contribute to this challenge.

## A.2 SOFTWARE FOR GENERATING THE DATASETS

Julia is used for the simulations Bezanson et al. (2017) and the dynamic simulations rely on the package DifferentialEquations.jl Rackauckas & Nie (2017). For simulating more realistic power grids in future work we recommend the additional use of NetworkDynamics.jl Lindner et al. (2021) and PowerDynamics.jl Plietzsch et al. (2021).

## A.3 SOFTWARE FOR TRAINING

The training is implemented in Pytorch (Paszke et al., 2019). For the graph handling and graph convolutional layers we rely on the additional library PyTorch Geometric (Fey & Lenssen, 2019). We use the SGD-optimizer and as loss function we use the mean squared error [1]. Furthermore `ray` (Moritz et al., 2018) is used for parallelizing the hyperparameter study.

## A.4 MODELLING DETAILS OF GENERATING THE DATASETES

In the detail of the modeling and the size of the dataset we attempt to strike a balance between relevance to real-world applications, computational tractability and conceptual simplicity. Therefore we employ the following criteria: (i) generate synthetic network topologies that mimic real world power grids; (ii) model the main dynamics of self-organised power flow and synchronization; (iii) minimize statistical and numerical errors with highly accurate simulations; (iv) to study out-of-distribution tasks and scale effects, consider grid sizes of different orders of magnitude.

The most important simplifications in comparison to real-world power grids are homogeneous edges, fixed magnitudes of sources/sinks and modeling all nodes by the swing equation. In contrast to our modeling, real power grid lines have different properties and there are more complex models for generators and loads. However, previous studies have shown that many interesting observations are still possible under our assumptions Nitzbon et al. (2017).

To investigate different topological properties of differently sized grids, we generate two datasets with either 20 or 100 nodes per grid, referred to as dataset20 and dataset100. To enable the training of complex models, both datasets consist of 10,000 graphs. Additionally, probabilistic dynamic stability values of a synthetic model of the Texan power grids are provided for evaluation purposes.

**Modeling of the power grids** This section covers the precise modeling of power grids used for the dataset generation and may be skipped. To generate realistic topologies we use the package `SyntheticNetworks`(Schultz et al., 2014b; Schultz, 2020). The sources and sinks are assigned randomly. For the dynamic simulations all nodes are represented by the $2^{nd}$-order-Kuramoto model (Kuramoto, 2005; Rodrigues et al., 2016), which is also called swing equation, see eq. (1). Using homogeneous coupling strength ($K$) can be interpreted as considering power grids that only have one type of power line and comparable distances between all nodes. This assumption is justifiable,

---

[1]corresponds to MSELoss in Pytorch

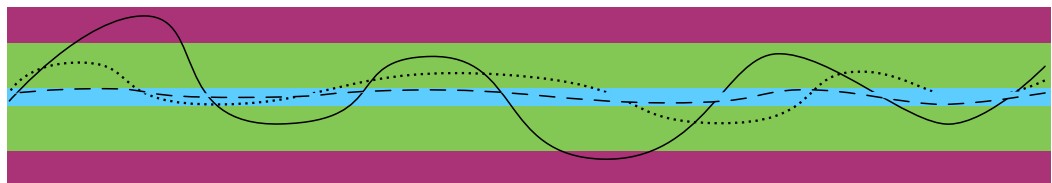

Figure 6: Identification of *troublemakers* based on trajectories. There are three spaces: blue for the space of perturbations, green for the *safe*-space and purple for the *trouble*-space. The dashed trajectory stays within the initial space of the perturbations, the dotted line leaves that space, but stays below the *trouble*-threshold and the solid line represents a trouble-maker-trajectory.

because in real power grids longer lines are built stronger, so the actual coupling does not scale as much with the length.

To estimate SNBS we rely on the approach presented in Nauck et al. (2022): "[F]or every node in a graph, $M = 10,000$ samples of perturbations per node are constructed by sampling a phase and frequency deviation from a uniform distribution with $(\phi, \dot{\phi}) \in [-\pi, \pi] \times [-15, 15]$ and adding them to the synchronized state. Each such single-node perturbation serves as an initial condition of a dynamic simulation of our power grid model, [cf. Equation (1)]. At $t = 500$ the integration is terminated and the outcome of the Bernoulli trial is derived from the final state. A simulation outcome is referred to as *stable* if at all nodes $\dot{\phi}_i < 0.1$. Otherwise it is referred to as *unstable*. The classification threshold of $0.1$ is chosen accounting for minor deviations due to numerical noise and slow convergence rates within a finite time-horizon."

To ensure the reliability of our results we try to minimize numerical and statistical errors: For the dynamical simulations a higher order Runge-Kutta methods with adaptive time stepping and low error tolerances is used. For the Monte-Carlo sampling, 10,000 simulations per node result in standard errors for our probabilistic measure SNBS of less than $\pm 0.01$.

Furthermore, we want to provide some numbers regarding the simulation time. The computation of a single perturbation in case of dataset20 takes .056 seconds, in case of dataset100 0.189 seconds and in case of the Texan power grid 239.97 seconds.

## A.5 IDENTIFICATION OF TROUBLEMAKERS

The idea is illustrated in Figure 6. The critical threshold is motivated by the real-world power grids. When certain operational bounds are violated, machines in the system switch of, potentially triggering failure cascades and large blackouts. Thus, there is a discontinuous difference between perturbations that attain a maximum deviation above the critical threshold and those that don't. The maximum limits for frequency deviations for the European grid are $+2$Hz or $-3$Hz. For our model we chose the symmetric critical threshold $\sim 2.4$Hz, corresponding to a maximum deviation of the angular velocity of $|\dot{\phi}| < 15$, both for simplicity and comparability to previous work.

**Selection of perturbations for the identification of troublemakers** If perturbations are amplified by more than a factor of 6, the desired space is left and we consider the trajectories as troublesome. If that occurs for at least one perturbation applied at a node, we consider that node a troublemaker, because such a strong amplification is a danger for safe grid operation. For the analysis of TM, we only consider perturbations within $(\dot{\phi}) \in [-2.5, 2.5]$. This reduces the minimum number of analyzed perturbations per node to 1,595 for dataset20, 1,569 for dataset100 and 129 for the Texan power grid. As mentioned in Section 2.1, the probability of rare events depends on the estimator and on the number of samples used for the estimation. As a consequence for dataset20 and dataset100 the rare even rate is $\gamma \approx 0.0013$ and for the Texan grid $\gamma \approx 0.015$. These values are derived from the Agresti-Coull approximation of a binomial confidence interval.

## A.6 EVALUATION OF THE PERFORMANCE USING DIFFERENT METRICS

For the regression, we use the coefficient of determination: $R^2 = 1 - \frac{mse(y,t)}{mse(t_{mean},t)}$, where $mse$ denotes the mean squared error, $y$ the output of the model, $t$ the target value and $t_{mean}$ the mean of all considered targets of the test dataset. $R^2$ captures the mean square error relative to a null model that predicts the mean of the test-dataset for all points. The $R^2$-score is used to measure the portion

of explained variance in a dataset. By design, a model that predicts the mean of SNBS per grids has $R^2 = 0$.

For the classification, we use F2-score which is based on recall and precision. The recall is defined by: Recall $= \frac{TP}{TP+FN}$, where $TP$ denotes true positives and $FN$ false negatives. The precision is defined by: precision $= \frac{TP}{TP+FP}$. This leads to the F2-score which is defined by: $F - 2 = \frac{5 \times precision \times recall}{4 \times precision + recall}$.

## A.7 HYPERPARAMETER OPTIMIZATION

We conduct hyperparameter studies in two steps. First, we optimize model properties such as the number of layers and channels as well as layer-specific parameters e.g. the number of stacks and internal layers in case of ArmaNets. For this optimization we use dataset20 and the SNBS task only. For all models we investigated the influence of different numbers of layers and the numbers of channel between multiple layers. We limit the model size to just above four million parameters, so we did not investigate the full presented space, but limited for example the number of channels when adding more layers. The resulting models have the following properties: ArmaNet has 3 layers and 189,048 parameters. GCNNet has 7 layers and 523,020 parameters. SAGENet has 8 layers and 728,869 parameters. TAGNet has 13 layers and 415,320 parameters.

Afterwards we optimize the learning rate, batch size and scheduler of the best models for dataset20 and dataset100 and the tasks SNBS/TM separately. Hence, our models are not optimized to perform well at the out-of distribution task. The best model from (Nauck et al., 2022) referred to as Ar-bench, is used as another baseline. It is a GNN model consisting of 1,050 parameters and based on 2 Arma-layers. The only adjustment to that model is the removal of the fully connected layer after the second Arma-Convolution and before applying the Sigmoid-layer, which improves the training.

## A.8 DETAILS OF THE TRAINING OF THE BENCHMARK MODELS

To reproduce the obtained results, more information regarding the training is provided in this section. Detailed information on the training as well as the computation time is shown in Table 3. In case of dataset20, a scheduler is not applied, in case of dataset100, schedulers are used for Ar-bench (stepLR), GCNNet (ReduceLROnPlateau). The default validation and test set batch size is 150. The validation and test batchsize for Ar-bench and ArmaNet3 is 500 in case of dataset20 and 100 for dataset100. The number of trained epochs differs, because the training is terminated in case of significant overfitting. Furthermore, different batch sizes have significant impact on the simulation time. Most of the training success occurs within the first 100 epochs, afterwards the improvements are relatively small.

Table 3: Properties of training models and regarding the training time, we train 5 seeds in parallel using one nVidia V100.

| name | number of epochs | | training time | | train batch size | | learning rate | |
|---|---|---|---|---|---|---|---|---|
| dataset | 20 | 100 | 20 (hours) | 100 (days) | 20 | 100 | 20 | 100 |
| Ar-bench | 1,000 | 800 | 26 | 4 | 200 | 12 | 0.914 | .300 |
| ArmaNet | 1,500 | 1,000 | 46 | 6 | 228 | 27 | 3.00 | 3.00 |
| GCNNet | 1,000 | 1000 | 29 | 5 | 19 | 79 | .307 | .286 |
| SAGENet | 300 | 1000 | 9 | 5 | 19 | 16 | 1.10 | 1.23 |
| TAGNet | 400 | 800 | 11 | 4 | 52 | 52 | 0.193 | .483 |

## A.9 DETAILED RESULTS OF TRAINING ON A SMALLER DATASET

To investigate the influence of available training data and to connect with previous work, we train all models on only 800 grids, from Nauck et al. (2022). The results are shown in Table 4.

Table 4: Performance after training on smaller training set. All models are trained on the same 800 grids as in Nauck et al. (2022), but evaluated on the newly introduced test set. The results are represented by $R^2$ score in %.

| model | tr20ev20 | tr100ev100 | tr20ev100 |
|---|---|---|---|
| Ar-bench | $46.54 \pm 2.378$ | $59.73 \pm 0.886$ | $31.75 \pm 1.204$ |
| ArmaNet | $70.35 \pm 1.226$ | $\mathbf{83.92} \pm 0.263$ | $55.84 \pm 0.598$ |
| GCNNet | $49.59 \pm 0.513$ | $61.18 \pm 1.663$ | $36.08 \pm 0.625$ |
| SAGENet | $50.15 \pm 0.255$ | $60.98 \pm 0.279$ | $39.89 \pm 0.089$ |
| TAGNet | $\mathbf{74.77} \pm 0.370$ | $82.21 \pm 0.017$ | $\mathbf{60.31} \pm 0.732$ |

### A.10  POOR PERFORMANCE OF GCN WHEN APPLYING TO THE TEXAN POWER GRID

The GCN model is not able to predict the dynamic stability for the Texan power grid. To understand this behaviour, we compare it to the ArmaNet model at the out-of-distribution task from dataset20 and dataset100 to predict SNBS of the Texan power grid. The scatter plots are shown in Figure 7. We can clearly see that the model is not able to predict lower values of SNBS correctly. The limited output of the GCNNet results in a bad performance in case of the distribution of the Texan power grid that has three modes. As a consequence, a model that predicts the mean of the distribution would achieve better performance. Furthermore, we provide the scatter plots of the GCNNet for the three tasks dataset20, dataset100 and tr20ev100 in Figure 8, that can be compared to Figure 4.

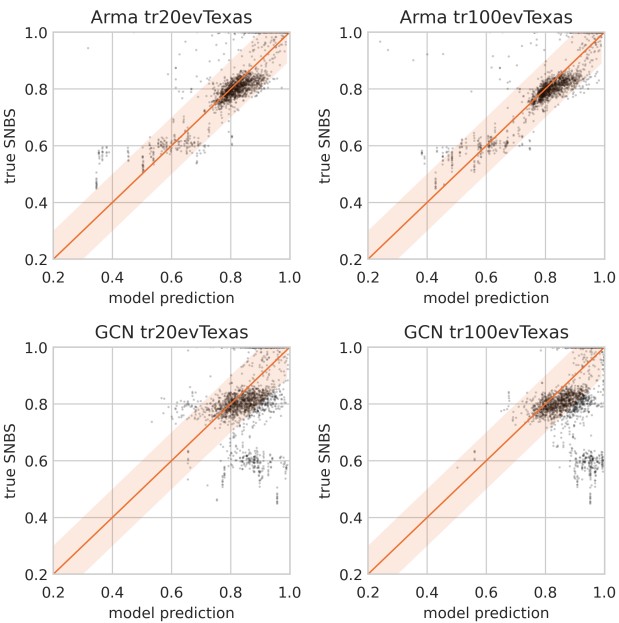

Figure 7: SNBS over predicted output of the Arma and GCN models for the out-of-distribution task to predict SNBS of the Texan power grid. The diagonal represents a perfect model ($R^2 = 1$), the banded region indicates prediction errors $\leq 0.1$. To account for the small number of nodes, a lower transparency is used in comparison to Figures 4 and 8.

### A.11  PERFORMANCE OF IDENTIFYING TROUBLEMAKERS

In the following sections we show the full results of predicting the troublemakers using classification and regression setup. We use the previously introduced GNN models (see section 3.1). For each regression and classification task we conduct another hyperparameter study to optimize the learning rates. The same inputs are used: adjacency matrix $A$, nodal power input $P$.

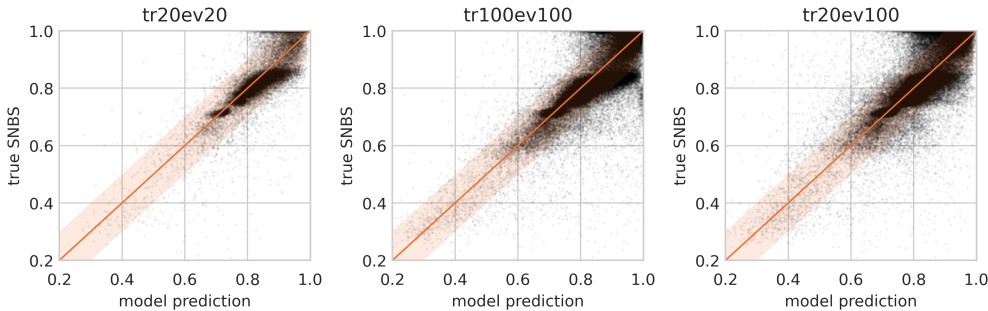

Figure 8: SNBS over predicted output of the GCNNet model for tr20ev20, tr100ev100 and tr20ev100. The diagonal represents a perfect model ($R^2 = 1$), the band indicates the region for accurate predictions based on a threshold of 0.1.

### A.11.1 CLASSIFICATION SETUP

For the classification setup, we report the F2-score in Table 5 and the recall in Table 6. The GNNs outperform the baselines in all tasks and especially TAGNet achieves a good performance. TAGNet achieves recalls of more than 93 % throughout all tasks, including the out-of-distribution generalizations, while still keeping F2-score above 82 %. For identifying troublemakers, a thresholded variant of the 'semi-analytic lower bound for survivability´ (Hellmann et al., 2016) is used as a further baseline. This bound can be directly computed from the input data and hence requires no training.

Table 5: TM prediction: F2-score in %. For MLP2, training on dataset100 was successful with only one of the five seeds. We report that result as over-estimation of a strong baseline. (*) Since the semi-analytic baseline does not require training, its performance is directly evaluated on the test set.

| model | tr20ev20 | tr100ev100 | tr20ev100 | tr20evTexas | tr100evTexas |
|---|---|---|---|---|---|
| Ar-bench | 74.31 ± 0.230 | 88.29 ± 0.331 | 77.08 ± 0.989 | 78.49 ± 0.790 | 83.33 ± 1.856 |
| ArmaNet | 86.85 ± 0.503 | 95.75 ± 0.049 | 84.94 ± 4.050 | 82.43 ± 1.747 | 85.14 ± 1.335 |
| TAGNet | **89.78** ± 0.079 | **96.36** ± 0.063 | **88.77** ± 0.309 | **82.78** ± 2.122 | **92.00** ± 1.585 |
| log | 44.74 | 73.47 | 5.59 | 21.66 | 55.81 |
| MLP1 | 58.06 ± 1.451 | 79.61 ± 0.001 | 72.68 ± 6.142 | 49.52 ± 9.990 | 86.02 ± 0.000 |
| MLP2 | 69.43 ± 0.724 | 91.60 | 41.11 ± 7.200 | 32.16 ± 12.338 | 73.22 |
| semi-analytic | 22.86* | 38.53* | 38.53* | 15.73* | 15.73* |

Table 6: TM prediction: recall in %. In case of MLP2, the training on dataset100 was successful with only two of the five seeds. We only report that result as over-estimation of a strong baseline.

| model | tr20 | | | tr100 | |
|---|---|---|---|---|---|
| | ev20 | ev100 | evTexas | ev100 | evTexas |
| Ar-bench | 86.30 ± 0.727 | 92.56 ± 0.798 | 83.33 ± 0.983 | 98.69 ± 0.654 | 98.04 ± 0.000 |
| ArmaNet | 88.85 ± 1.019 | 96.85 ± 0.272 | 86.74 ± 5.455 | 90.20 ± 2.264 | 98.69 ± 0.654 |
| TAGNet | **93.84** ± 0.039 | **97.16** ± 0.120 | **96.10** ± 0.590 | **99.35** ± 0.654 | **100.00** ± 0.000 |
| logreg | 86.69 | 91.82 | 95.32 | 89.04 | 94.12 |
| MLP1 | 76.44 ± 0.312 | 90.70 ± 0.000 | 68.98 ± 6.874 | 44.44 ± 9.628 | 94.12 ± 0.000 |
| MLP2 | 75.85 ± 0.513 | 89.86 | 36.08 ± 6.902 | 28.10 ± 11.563 | 68.63 |
| semi-analytic | 92.70* | **97.77*** | 98.04* | 97.77* | 98.04* |

### A.11.2 REGRESSION SETUP

Besides introducing two classes for predicting troublemakers, it is also possible to directly predict the maximum frequency deviation per node (fig. 2). The results are shown in table 7. The predictions of the regression can be complemented by applying a thresholding afterwards to categorize nodes

as troublemakers. After applying the thresholding, the results can again be evaluated using F2-score (table 8) and recall table 9.

Table 7: Results of predicting maxiumum frequency devations represented by $R^2$ score in %. In case of MLP2, the training on dataset100 was successful with only two of the five seeds. We only report the best result as over-estimation of a strong baseline.

| model | tr20ev20 | tr100ev100 | tr20ev100 | tr20evTexas | tr100evTexas |
|---|---|---|---|---|---|
| Ar-bench | $87.45_{\pm 0.132}$ | $92.73_{\pm 0.132}$ | $89.21_{\pm 0.226}$ | $77.76_{\pm 0.614}$ | $82.45_{\pm 1.302}$ |
| ArmaNet | $\mathbf{96.59}_{\pm 0.074}$ | $97.70_{\pm 0.096}$ | $\mathbf{95.70}_{\pm 0.150}$ | $\mathbf{86.73}_{\pm 1.432}$ | $\mathbf{90.70}_{\pm 1.216}$ |
| TAGNet | $96.30_{\pm 0.258}$ | $\mathbf{97.99}_{\pm 0.062}$ | $93.68_{\pm 0.270}$ | $81.88_{\pm 0.477}$ | $84.81_{\pm 0.790}$ |
| linreg | $83.87$ | $87.77$ | $86.60$ | $80.85$ | $78.12$ |
| MLP1 | $90.55_{\pm 0.025}$ | $92.54_{\pm 0.228}$ | $82.21_{\pm 1.256}$ | $54.64_{\pm 16.32}$ | $52.71_{\pm 17.46}$ |
| MLP2 | $89.95_{\pm 0.144}$ | $92.30$ | $81.85_{\pm 1.454}$ | $60.83_{\pm 7.764}$ | $83.82$ |

Table 8: Results of predicting troublemakers using regression and thresholding represented by F2-score in %. In case of MLP2, the training on dataset100 was successful with only two of the five seeds. We only report the best result as over-estimation of a strong baseline.

| model | tr20ev20 | tr100ev100 | tr20ev100 | tr20evTexas | tr100evTexas |
|---|---|---|---|---|---|
| Ar-bench | $44.49_{\pm 0.018}$ | $78.66_{\pm 1.639}$ | $58.28_{\pm 1.302}$ | $85.24_{\pm 0.176}$ | $89.15_{\pm 1.150}$ |
| ArmaNet | $81.68_{\pm 0.509}$ | $88.29_{\pm 5.898}$ | $\mathbf{91.49}_{\pm 0.740}$ | $87.15_{\pm 2.922}$ | $91.86_{\pm 1.058}$ |
| TAGNet | $\mathbf{84.55}_{\pm 1.136}$ | $\mathbf{95.13}_{\pm 0.316}$ | $90.20_{\pm 1.624}$ | $\mathbf{90.18}_{\pm 1.645}$ | $\mathbf{95.48}_{\pm 0.562}$ |
| linreg | $72.69$ | $91.51$ | $91.32$ | $73.22$ | $93.75$ |
| MLP1 | $74.33_{\pm 0.074}$ | $91.60_{\pm 0.006}$ | $81.60_{\pm 2.155}$ | $75.17_{\pm 2.837}$ | $41.96_{\pm 16.52}$ |
| MLP2 | $74.38_{\pm 0.036}$ | $91.61$ | $77.46_{\pm 7.020}$ | $68.89_{\pm 5.248}$ | $93.75$ |

Table 9: Results of predicting troublemakers using regression and thresholding represented by recall in %. In case of MLP2, the training on dataset100 was successful with only two of the five seeds. We only report the best result as over-estimation of a strong baseline.

| model | tr20ev20 | tr100ev100 | tr20ev100 | tr20evTexas | tr100evTexas |
|---|---|---|---|---|---|
| Ar-bench | $39.85_{\pm 1.792}$ | $75.28_{\pm 1.892}$ | $53.29_{\pm 1.476}$ | $87.58_{\pm 0.654}$ | $93.46_{\pm 1.307}$ |
| ArmaNet | $78.80_{\pm 0.557}$ | $86.37_{\pm 6.948}$ | $91.60_{\pm 1.033}$ | $88.89_{\pm 3.458}$ | $95.42_{\pm 0.654}$ |
| TAGNet | $\mathbf{82.65}_{\pm 1.337}$ | $\mathbf{94.47}_{\pm 0.433}$ | $\mathbf{91.66}_{\pm 2.38}$ | $\mathbf{94.77}_{\pm 1.729}$ | $\mathbf{96.73}_{\pm 0.654}$ |
| linreg | $70.55$ | $89.75$ | $68.63$ | $89.54$ | $94.12$ |
| MLP1 | $69.89_{\pm 0.079}$ | $89.87_{\pm 0.011}$ | $79.80_{\pm 3.995}$ | $86.93_{\pm 7.190}$ | $47.06_{\pm 24.49}$ |
| MLP2 | $69.93_{\pm 0.039}$ | $89.88$ | $73.68_{\pm 7.778}$ | $64.05_{\pm 5.584}$ | $94.12$ |

