# OpenReview forum: "Towards predicting dynamic stability of power grids with Graph Neural Networks"
_ICLR.cc/2023/Conference — Submitted to ICLR 2023_

### Official Review · Reviewer_iZuM · 2022-10-19

**Confidence:** 3
**Correctness:** 3
**Technical Novelty And Significance:** 3
**Empirical Novelty And Significance:** 3
**Recommendation:** 5

**Clarity, Quality, Novelty And Reproducibility:**

The originality of this paper is good, since it provides a new dataset about a significant real-world problem with detailed description of the generation of data so that it is trustful.

The novelty of this paper is lacking, but it is just because of the output of this paper. It provides a larger dataset than the previous ones, rather than propose a new research problem.

The clarity of this paper is good enough, except the description of identification of trouble-maker nodes.

The reproducibility of this paper is good, since it provides the experimental details to reproduce the results and the authors promise that they will open-source the dataset and codes afterwards.

The general quality of this paper is above the average in my view.



**Strength And Weaknesses:**

## Strength:
1. This paper is generally well written and the background is sufficient to let the people who is not an expert in power systems understand the task.
2. The details of generating data are provided.
3. The figures are concise to show the ideas.
4. The anlysis of experimental results are sufficient and solid.

## Weaknesses:
1. No significant novelty appeared in this paper.
2. The method to identify the trouble-maker nodes is not clearly described at least from the main part of paper. I suggest the authors can merge the related contents in the appendix and rewrite it using compact words.
3. The authors claim that the trouble-maker identification is a classification task. However, in my view it is just a criterion rather than a classification task that need the ML model to conduct.


**Summary Of The Paper:**

This paper introduces a new dataset of probabilistic dynamic stability of synthetic power grids which is much larger than the existing datasets. Moreover, it also includes a Texan power grid model alongside with the dataset. Second, this paper also proposes a new method to identify the nodes which could be the one leading to the unstability. Finally, the papers evaluate several GNN baselines on the proposed benchmark and made a complete analysis. In the experimental results, the paper shows the potential to use the GNN model trained with the offline data to conduct the prediction on the real-world power grid.

**Summary Of The Review:**

This paper provides a new benchmark dataset about a significant real-world problem and it has shown the evaluation with several baselines which could provide some lessons to the following people. The background of the problem is sufficient. The description of the identification of trouble-maker nodes needs to be clarified, which may change my decision.

---

> ### Author Response · Authors · 2022-11-08
> **Reply to Reviewer iZuM**
>
> We thank reviewer iZuM for the valuable feedback and are pleased that the reviewer appreciates the originality of the submission. We address the individual concerns below.
>
> > *The originality of this paper is good, since it provides a new dataset about a significant real-world problem with detailed description of the generation of data so that it is trustful.*
> >
> > *The novelty of this paper is lacking, but it is just because of the output of this paper. It provides a larger dataset than the previous ones, rather than propose a new research problem.*
>
> In our opinion there are also several points that are not just original but also novel:
>
> 1. We provide new and much larger datasets. The increased size of the dataset allows for new insights in the applicability of GNNs.
> 2. The new GNN models outperform previously published approaches including smaller GNNs and conventional baselines. Interestingly, the previously published GNNs are worse than some conventional baselines.
> 3. To show the potential for future applications, we add a real-sized Texan power grid model and observe the general applicability of our approach.
> 4. We introduce a new target: Trouble makers, which are nodes that highly amplify perturbations and hence are a danger to grid stability.
>
> > The method to identify the trouble-maker nodes is not clearly described
> >
> > The authors claim that the trouble-maker identification is a classification task. However, in my view it is just a criterion rather than a classification task that need the ML model to conduct.
>
> We hope the reply to all reviewers (1-4) clarifies the idea, its relevance and the ML task we perform. We are happy to provide more detail if desired.

---

> > ### Comment · Reviewer_iZuM · 2022-11-18
> > **Thank you for your response**
> >
> > Dear Authors,
> >
> > I have read the response. I can only say the points you mentioned are findings, but I don't deny it is a good submission. I will discuss with other reviewers about the paper to give a final decision.

---

> > > ### Author Response · Authors · 2022-12-08
> > > **Response**
> > >
> > > We thank reviewer iZuM for considering the paper a good submission. In case new questions came or come up during the internal discussion, we would be happy to address them.

---

### Official Review · Reviewer_SqnB · 2022-10-24

**Confidence:** 3
**Correctness:** 3
**Technical Novelty And Significance:** 3
**Empirical Novelty And Significance:** 4
**Recommendation:** 6

**Clarity, Quality, Novelty And Reproducibility:**

This work is novel as it introduces a new synthetic dataset and extends previous models.
I think the clarity and quality of the writing could be improved by carefully checking the typos (a few entire sentences do not make sense, even in the abstract)

**Strength And Weaknesses:**

I am unfortunately not super knowledgeable about power-grids but enjoyed reading this interesting paper.

I believe that the strength of this paper lies in the introduction of a new dataset of higher complexity than previous existing datasets. I think this is very important for evaluating and training models.
The other strength of the proposed model is the computational efficiency and scalability (1,800 times faster for grids of size 100 and 4,6 ×107 times faster for the synthetic Texan power grid).

The fact that GNNs can generalize from a small dataset to a bigger one is very compelling, but the datasets are very similar, and generated using the same modeling strategy. I am not surprised that a GNN can generalize well in this setting, since what it learns from dataset20 will hold in dataset100. I think that this result is overstated by the authors. I might not understand the differences between dataset20 and dataset100 well enough to understand the depth of this statement. I'd appreciate if the authors could explain this to me in more details.

I'm also not sure that it makes sense to flag a node as trouble maker if small disturbances are amplified by a factor of at least 6. This seems a bit arbitrary. Is there any reason coming from analysis of real datasets for this?
This is then used for a classification task whereas I think a regression task would be more appropriate. For example, predicting the amplification factor rather than predicting if the factor is above or below 6. Also, reporting the uncertainty should be interesting. Is the model more confident on nodes with higher amplification factor? To use in a real setting, we would want the model to have a good uncertainty representation.

One thing I didn't understand is why, in figure 3, the texan grid has 97% of stable nodes although the distribution of SNBS seems to give much less probability for high SNBS (around 1). I'd appreciate some clarification here.

In figure 5, the model seems to work very well for low SNBS but have skewed predictions for high SNBS. Would it be possible to fix this using some non-linearity?

Overall, there are lots of typos that make the paper a bit hard to read. For example in the abstract:
We show that large GNNs outperform GNNs from previous work as well as as handcrafted graph features and semi-analytic approximations.





**Summary Of The Paper:**

The authors introduce new datasets of dynamic stability of synthetic power grids.
They show that large Graph Neural Networks GNNs outperform GNNs from previous work at predicting single-node basin stability.They demonstrate that GNNs can be used to identify trouble maker-nodes in the power grids and show that GNNs trained on small grids can perform accurately on a large synthetic Texan power grid model.

**Summary Of The Review:**

I am not very confident, and overall enjoyed the paper.
The new dataset is a big contribution, but I would like some clarification on the tasks that are used to evaluate the model (trouble node classification, SNBS prediction) as I'm not sure they fully support the claims made in the paper.

---

> ### Author Response · Authors · 2022-11-08
> **Reply to Reviewer SqnB**
>
> We thank the reviewer for the useful comments. We respond to the concerns below.
>
> The reviewer questions if the generalization from smaller to larger grids is challenging.
>
> > I am not surprised that a GNN can generalize well in this setting, since what it learns from dataset20 will hold in dataset100.
> >
> > the datasets are very similar, and generated using the same modeling strategy.
>
> We have outlined in the joint reply (3) why this is a challenging problem.
>
> > I'm also not sure that it makes sense to flag a node as trouble maker if small disturbances are amplified by a factor of at least 6.
> >
> > I think a regression task would be more appropriate.
>
> Thanks for your interesting idea, please see Reply to all reviewers (4).
>
> > I didn't understand [...] in figure 3, the texan grid has 97% of stable nodes
>
> The figure may be confusing and will be changed to make it clearer. The histogram shows SNBS, but the bar below shows SURV and the number of trouble makers. Since SNBS considers the final state of the trajectory, whereas SURV considers the entire transient trajectory, it is impossible to draw conclusions for SURV out of the histogram showing SNBS. Perturbations can survive (SURV) but not end in the basin and hence their SNBS be unstable. Real power grids require both types of stability.
>
> > the model seems to work very well for low SNBS but have skewed predictions for high SNBS
>
> That is an interesting idea. Indeed, there appear to be multiple classes of nodes which the GNNs seem to be unable to distinguish, considering the histograms in Figure 3 and multiple horizontal structures in Figure 5. Since these classes of nodes overlap in the predictions, we do not expect non-linearities to be able to disentangle them. Nevertheless, we will try that and provide updates in the next days.
>
> > typos that make the paper a bit hard to read. For example in the abstract: We show that large GNNs outperform GNNs from previous work as well as as handcrafted graph features and semi-analytic approximations.
>
> We are very sorry for the typos and working on improving the text. To improve readability, we changed the abstract: "We show that large GNNs outperform GNNs from previous work, and that the new GNNs also outperform handcrafted graph features as well as semi-analytic approximations.
>
> > I would like some clarification on the tasks that are used to evaluate the model (trouble node classification, SNBS prediction)
>
> We hope we have addressed this satisfactorily in the joint replies. If not, we are happy to provide more details.

---

> > ### Author Response · Authors · 2022-11-16
> > **Update on concerns and ideas**
> >
> > > in figure 3, the Texan grid has 97% of stable nodes although the distribution of SNBS seems to give much less probability for high SNBS
> >
> > We updated the figure (now figure 2) to include the distribution of maximum frequency deviations which is used to identify troublemakers. We clarified that the number of 97% stable nodes refers to the troublemakers and not to SNBS. We also clarified the difference between transient stability and asymptotic stability.
> >
> > > fix this using some non-linearity
> >
> > We tried adding non-linearities (prior and after training) by adding an additional sigmoid or tanh function, but that resulted in much lower performance. If the reviewers have specific ideas of applying further non-linearities, we are happy to try them out.
> >
> >
> > > I think a regression task would be more appropriate.
> >
> > The following tables shows the results of successfully identifying troublemakers using regression represented by F2-score. More information can be found in the paper in 3.3 Identification of troublemakers and Appendix 11.2
> >
> > |model |tr20ev20 | tr100ev100| tr20ev100| tr20evTexas| tr100evTexas|
> > |---|---|---|---|---|---|
> > |ArmaNet|81.68 |88.29|91.49|87.15|91.86|
> > |TAGNet|84.55|95.13|90.20|90.18|95.48|
> > |linreg|72.69|91.51|91.32|73.22|93.75|
> > |MLP1|74.33|91.60|81.60|75.17|41.96|
> > |MLP2|74.38|91.61|77.46|38.89|93.75|

---

> > > ### Comment · Reviewer_SqnB · 2022-12-07
> > > **Re: Reply to Reviewer SqnB**
> > >
> > > I thank the authors for addressing my comments and responding to my questions.
> > > I believe that the regression results and the update of Figure 3 have made the paper of better quality and have thus raised my score.
> > >
> > > I agree with reviewer Wgck and I think this paper can be considered as a reasonable dataset contribution.

---

> > > > ### Author Response · Authors · 2022-12-08
> > > > **Response**
> > > >
> > > > We thank reviewer SqnB for the kind reply. We are happy that we successfully answered the questions and addressed the comments in the revision. If further questions arise, we'd be happy to address them too.

---

### Official Review · Reviewer_wgck · 2022-10-30

**Confidence:** 4
**Correctness:** 4
**Technical Novelty And Significance:** 2
**Empirical Novelty And Significance:** 3
**Recommendation:** 6

**Clarity, Quality, Novelty And Reproducibility:**

While there is great potential in this paper, I think there is unfortunately great room for improvement in terms of clarity and reproducibility. The novelty is in that dynamic stability datasets of the presented size do not exist in the literature. The quality of the dataset is likely good, but it is difficult to tell from the writeup.

**Strength And Weaknesses:**

Strengths:
* The proposed dataset characterizes an important problem. Specifically, the problem of dynamic stability is indeed increasingly important to address as power grids integrate larger proportions of time-varying renewable energy.

Weaknesses:
* The dataset structure is not clearly described, and the benchmark task (what are the inputs, what are the outputs, and what is the metric of success) is not cleanly defined. This information is implicit within the text and citations, but needs to be laid out much more clearly and explicitly to enable understanding of the specific benchmark presented - e.g., I would have expected to see some of the relevant equations laid out in the paper.
* The setup of the baseline GNNs is also not clearly enough described. For instance, the text describes that the GNNs are given "an adjacency matrix representing the topology of power grids and a binary feature vector representing sources and sinks," without additional detail - however, for a dataset paper, those inputs should be clearly spelled out.
* Given that this is a machine learning audience as opposed to a power systems audience, the introduction needs to be made much more accessible. For instance, terms/phrases such as "have less inertia," "contingencies," and "self-organized synchronization mechanism" may not be immediately accessible to an ML audience. In addition, the word "transformers" should likely be clarified as "power transformers," due to the different default meaning of this term in the deep learning literature.
* Since all the data is synthetically generated, it is not clear to me why this is presented as only a static dataset, rather than the data generation code also being shared. See, e.g., [1] for an example of a dataset paper that does it this way.

[1] Joswig-Jones, Trager, Kyri Baker, and Ahmed S. Zamzam. "OPF-Learn: An open-source framework for creating representative AC optimal power flow datasets." 2022 IEEE Power & Energy Society Innovative Smart Grid Technologies Conference (ISGT). IEEE, 2022.

**Summary Of The Paper:**

This paper presents a dataset containing synthetic models of power grids combined with the statistical results of dynamic simulations (quantifying single-node basin stability and survivability - abbreviated SNBS and SURV). The authors train a baseline GNN to show initial efficacy of the task of predicting SNBS and SURV, as well as of transferring the learned model to larger power grids.

**Summary Of The Review:**

The proposed dataset characterizes an important problem. However, the writeup does not clearly define the dataset/benchmark task or the baseline method. In addition, it would likely make more sense to present this as a software package for generating data, _alongside_ a fixed dataset, rather than just as a fixed dataset.

If the authors are able to make significant revisions to the writeup during the revision period, I would potentially be willing to significantly increase my score, given that my issues are (likely) with the presentation rather than the content of the work (at least, based on what I'm able to assess from the current writeup). In general, I think dataset papers are extremely valuable, particularly those dealing with climate/energy problems, but unfortunately the current writeup is not of high enough quality for this reviewer to recommend acceptance without such significant revisions.

---

> ### Author Response · Authors · 2022-11-08
> **Reply to Reviewer wgck**
>
> We thank reviewer wgck for the questions and suggestions. We are implementing the suggestions for the updated manuscript and address the identified weaknesses. Below, we respond to the specific concerns.
>
> > *The dataset structure is not clearly described, and the benchmark task [...] is not cleanly defined.*
> >
> > *expected to see some of the relevant equations laid out in the paper*
> >
> > *setup of the baseline GNNs is also not clearly enough described*
>
> The procedure of generating the datasets and the setup for the training using GNNs is explained in the reply to all Reviewers (1-3). We are happy to go more into detail if desired.
>
> > terms/phrases such as "have less inertia", "contingencies" and "self-organized synchronization mechanism" may not be immediately accessible to an ML audience
>
> We appreciate the great suggestions. We will add explanations on the meaning of inertia, contingencies and *self-organized synchronization mechanism and* replaced transformers by power transformers.\
> \
> Inertia is the reaction of a nodes frequency to a power imbalance. In classical power plants this really is the rotational inertia of the generator, which absorbs instant shocks in the power balance.
>
> Contingencies are major faults or problems that perturb the stable operating state and function of the power grid.
>
> The self-organized synchronization is the ability of all the generators to establish a joint frequency without external signal. It results from an interplay of droop control, inertia and power flow that is captured by the Swing equation given in the joint reply (1).
>
> > it is not clear to me why this is presented as only a static dataset, rather than the data generation code also being shared.
>
> All code will be made available upon publication, see Reply to all Reviewers (5. Code availability)

---

> > ### Comment · Reviewer_wgck · 2022-11-28
> > **Response**
> >
> > Thanks to the authors for their significant revision of the manuscript. My major concerns have been addressed, and I have raised my score accordingly.
> >
> > Please note that while this paper does not propose a novel research problem in the traditional sense (Reviewer iZuM), I do think the paper makes the previous dataset more amenable to a machine learning workflow (by increasing its size, training baseline models, etc.) and adds an additional task regarding identifying troublemakers (which can be viewed in some sense as a novel research problem). As such, I do think this paper can be considered as a reasonable dataset contribution.

---

> > > ### Author Response · Authors · 2022-11-28
> > > **Response**
> > >
> > > We thank reviewer wgck for the kind words. We are happy that our revision successfully addressed the raised concerns. If anything remains unclear, we'd be happy to address that, too.

---

### Author Response · Authors · 2022-11-08
**Reply to all Reviewers Sections 1-2**

We would like to thank all reviewers (wgck, SqnB, iZuM) for their time and their careful reading of the manuscript. We appreciate that they find our paper deals with an interesting and important challenge and we are glad that all reviewers highlighted the strengths of our dataset and paper. The reviewers underline the contribution: *"proposed dataset characterizes an important problem"* (wgck), *"novelty is in that dynamic stability datasets of the presented size do not exist"* (wgck), "*originality of this paper is good*" (iZuM), the soundness: *"enjoyed reading this interesting paper"* (SqnB)*,* "*This paper is generally well written"* (iZuM), *"the background is sufficient to let the people who is not an expert in power systems understand the task."* (iZuM) and implementation: *"extends previous models"* (SqnB), *"analysis of experimental results are sufficient and solid"* (iZuM).

Below we first address common concerns of the reviewers regarding the description of the datasets and tasks, the question of trouble maker identification, the precise challenge in transferring results from dataset20 to dataset100 and the Texas topology as well as the availability of the code. All improved explanations will also be worked into a revision of the manuscript.

## 1. The model to generate the datasets

The model underlying the dataset is described in Section 2, A4, A5. We will incorporate mathematical details in the main part of the manuscript. As it is relevant to the following discussion and to answer the questions on mathematical details of the task, we summarize them again and add relevant details to address the concerns.

The power grid simulations are based on solving coupled differential equations, which describe the dynamics per node. Power grids are represented by their topology via adjacency matrix A, the properties of the nodes and edges. In our case, we use homogeneous edges and the nodes are considered to be producers or consumers. The relevant dynamic quantity are the phase variables at the nodes and we use $\\phi\_i$ that represents the deviation of the local AC voltage from an arbitrary 50/60Hz reference.

The following model can describe the self-organized synchronization and stabilization of the active power flow:

$$ \\ddot \\phi\_i = P\_i - \\alpha \\dot \\phi\_i - K \\sum\_j A\_{ij} sin(\\phi\_i - \\phi\_j) $$

Synchronous operation requires $0 = \\ddot \\phi\_i = \\dot \\phi\_i$, that is a fixed offset and no frequency deviation. The fixed point equation $P\_i = K \\sum\_j A\_{ij} sin(\\phi^\*\_i - \\phi^\*\_j)$ is the power flow equation on a grid with topology given by the adjacency matrix $A$. We assume homogeneous coupling $K = 8$, and that all nodes participate in the control for power imbalances equally with homogeneous damping coefficient $\\alpha = 0.1$. These assumptions help to isolate the effect of the coupling topology encoded in $A$ on the dynamics.

Using available high performance ODE solvers, the entire grids trajectories starting from any given initial condition can be generated.

## 2. The task of predicting the dynamics using GNNs

Dynamic stability is crucial for the power grid to avoid blackouts, but difficult to measure. It has become state of the art to therefore use probabilistic metrics such as single-node basin stability (SNBS) and survivability (SURV). Each node is perturbed in turn leading to one distribution of initial conditions $\\rho\_i$ per node $i$. Writing $p(y|x \\sim \\rho) = \\int p(y|x) \\rho(x)$ for the compound distribution of $p$ with $\\rho$ we have

$$ SNBS\_i = p(\\phi(\\infty) = \\phi^\* | \\phi(0), \\dot \\phi(0) \\sim \\rho\_i) $$.

In words, SNBS is the probability that the entire system returns to the fixed point after the perturbation. A complementary stability measure is the SURV,

$$ SURV\_i = p(\\max\_{t, i} \\dot \\phi\_i(t) < \\text{ critical threshold } | \\phi(0), \\dot \\phi(0) \\sim \\rho\_i), $$

that is, the probability that the maximum frequency deviation of the entire system stays below a critical threshold. We estimate these probabilities by simulating 10.000 trajectories per node.

The goal of our work is the usage of GNNs to replace these expensive simulations. To that end, we train GNNs to learn the graph function $ (P, A) \\rightarrow SNBS $ and $ (P, A) \\rightarrow SURV $.

The underlying simulations to generate the datasets are feasible for anyone with some domain knowledge, but the composition of entire datasets require significant amounts of computational resources. For the datasets with 20,000 grids and the Texan power grid, the simulations take roughly 700,000 CPU hours. Thus an important contribution is to publish a full dataset to enable groups that have less computational resources to work on this important problem.

---

> ### Author Response · Authors · 2022-11-08
> **Reply to all Reviewers Sections 3-4**
>
> ## 3. The challenge of transfer from small to large grids
>
> The node wise features, SNBS and SURV, measure the reaction of the entire coupled grid to a localized perturbation. There is no a priori reason to assume that the 40 dimensional dynamical system on a 20 node grid will react the same as a 200 dimensional system on a 100 node grid. The performance of GNNs in general and the out-of-distribution generalization capabilities from small dynamical systems to larger ones in particular are quite surprising from a dynamical systems perspective. It was not clear that any generalization would be feasible.
>
> There are several empirical results that support that intuition in the submission.
>
> First evidence can be found in Figure 3, where the Texan power grid actually does show a different histogram. Compared to the smaller topologies, a third mode appears, indicating different dynamical effects occur in the larger network despite the same dynamic modelling method being used. Certain topological structures relevant to the dynamics of larger grids might simply be too large to exist in smaller grids. Additionally, the histograms for dataset20 and dataset100 also show significant differences.
>
> Further evidence for that can be found in the lack of generalization performance of the network-measure baselines. Several previous investigations relied on network science only, and they were not able to generalize well [Schultz Detours]. If the global structure played no role in the dynamical response, one would expect the baselines to generalize well.
>
> Finally, GNNs and MLPs perform better on tr100evTexas than on tr20evTexas. If the size did not have any impact, it should not matter whether models are trained on grids of size 20 or 100.
>
> Since the computational costs scale at least quadratically with the size of grids, out-of-distribution generalization might be one of the keys for real-world applications. GNN models could be trained on (sufficiently large) subgraphs and still be applied to entire real-world power grids.
>
> [Schultz Detours] Schultz et al. “Detours around basin stability in power networks.” New Journal of Physics, and private communication
>
> ## 4. Identification of Trouble makers
>
> The reviewers highlighted that the trouble makers may require additional explanation and motivation, which we are happy to provide below. SNBS deals with asymptotic stability whereas survivability (SURV) measures whether transient frequency deviations pass a critical threshold. This is motivated by real world constraints in power grids. When certain operational bounds are violated, machines in the system switch of, potentially triggering failure cascades and large blackouts. Thus, there is a discontinuous difference between perturbations that attain a maximum deviation above the threshold and those that don't. The maximum limits for deviations of the European grid frequency [DIN EN] are $+2$ Hz or $-3$ Hz. For our model we chose the symmetric critical threshold $2.4$Hz for simplicity and comparability to previous work.\
> \
> The usual definition of $\\rho\_i$ (see Reply 2) for SURV fills the whole state space region up to the critical threshold. When building the dataset, we observed that there are some few nodes for which a much narrower distribution of initial perturbations $\\rho^{narrow}\_i$ can nevertheless lead to frequency deviations that are sufficiently amplified to reach the threshold. We dubbed these nodes trouble makers and tuned the narrower distribution to focus on the worst offenders. This lead to initial perturbations confined to $\\pm 0.4$Hz, leading to an interesting amplification factor of 6 (.4 Hz x 6 = 2.4 Hz). We chose these nodes as a new target for our investigation:
>
> $ TM\_i = 1 $ if $ SURV\_i^{narrow} < 1 $ and 0 otherwise.
>
> The observation of large amplifications and the target function are both of practical importance and novel as far as we are aware.
>
> ### Identification of trouble makers using regression
>
> As noted by the reviewers, this task could be decomposed into a regression task and a thresholding. Directly estimating the maximum might be too noisy, but we could either try to learn $SURV\_i^{narrow}$ directly or try to learn the 99th percentile of the maximum frequency deviation:\
> \
> $$ P\_{99}[\\max\_{t, i} \\dot \\phi\_i(t) < \\text{ critical threshold } | \\phi(0), \\dot \\phi(0) \\sim \\rho^{narrow}\_i] $$
>
> and then apply the thresholding afterwards. We think this is an interesting way of framing the question and will look into whether this provides additional insights while revising the manuscript. We are moderately skeptical that this will work as well as directly predicting $TM\_i$ because $TM\_i$ focuses on the extreme ends of the SURV distribution, and a regression task would need to be carefully crafted to accurately capture this extreme end.
>
> [DIN EN] DIN EN 50160 Voltage characteristics of electricity supplied by public distribution networks

---

> > ### Author Response · Authors · 2022-11-08
> > **Reply to all Reviewers Section 5**
> >
> > ## 5. Code availability
> >
> > The code to generate the dataset is part of the submission. The dataset is meant to be reproducible and extendable. Unfortunately, we forgot to add the code for dataset generation to the google drive, but uploaded the scripts to generate synthetic grids and conducting the dynamical simulations to the Drive (<https://drive.google.com/file/d/19irz3CIuu8hI4Gz-VCWi84-YV>[\\\_6IllFh/view?usp=sharing](https://drive.google.com/file/d/19irz3CIuu8hI4Gz-VCWi84-YV%5C_6IllFh/view?usp=sharing)). We will provide all code in its current state on Zenodo (including the trained models and code to generate all figures and make it fully reproducible). Additionally, we will set up a GitHub repository to enable collaborative working and forks to reuse parts of the code for different problems. We are also happy to share additional code now if it is helpful for reviewing.

---

### Author Response · Authors · 2022-11-16
**Revision of the manuscript**

We thank the reviewers for their initial feedback and suggestions. We uploaded a revised version of the manuscript. The main changes are:

* Rewrite of parts of the introduction to include several explanations regarding power grid terms to make the article even more accessible to the ML community.
* The introduction now includes background information on the modeling approach and computation of dynamic stability, including the relevant equations.
* Figure 1 has been moved to the Appendix.
* Complete rewrite of the section introducing the concept of the troublemakers and motivation of the amplication factor 6.
* New section:  "Structure of datasets", where we show what data is included in the dataset and how it is structured. Figure 2 (formerly 3) now contains updated information on the troublemakers.
* Regarding the setup of the baseline GNNs, we updated the section "Experimental setup" including figure 3 (formerly figure 4) to clarify the inputs and outputs.
* The prediction of the troublemakers includes new results. Considering the suggestion by reviewer SqnB we also train the models using a regression setup and provide those results. In many cases this setup works even better than the classification.

Hopefully this revised version and our previous replies address all of the raised concerns and we are looking forward to the discussion.

---

### Decision · Program_Chairs · 2023-01-20

**Decision:**

Reject

**Justification For Why Not Higher Score:**

See above metareview.

**Justification For Why Not Lower Score:**

N/A

**Metareview: Summary, Strengths And Weaknesses:**

The reviewers were split about this paper and did not come to a consensus: on one hand they appreciated the importance of releasing datasets/benchmarks to the ICLR community (especially in the climate/energy space), on the other hand they were confused why the data was presented as static given that the authors have designed a simulator and argued that the description was not very accessible to an ML audience and is missing important details. After going through the paper and the discussion I have decided to vote to reject for the following reason: because this a dataset paper, presentation and empirical analysis should be held to a higher standard than a paper that also proposes new methodology. Even with the changes made by the authors in response to reviewer comments, the paper is not easy to access for a general ML reader. For example, the first five paragraphs of the introduction, while interesting, are overly detailed to an ML audience that is developing new models for the dynamic stability prediction problem. Much of it is historical background which can be condensed, or highly technical information which requires very specialized knowledge to parse. Mathematical notation is also often confusing, e.g., the authors use $p(y | x \sim \rho)$ to mean $p(y|x)$ integrated over distribution $\rho(x)$, instead of the standard $\mathbb{E}_{\rho(x)}[p(y|x)]$. In another example $\phi(\infty)$ is used, which I believe means $\phi$ at the final timestep, but prior to this $\phi_i$ is defined as a term indexed only by nodes $i$, its extension to a time-dependent function is never stated, but is assumed to be known in equation (3). Further the analysis is limited: e.g., Figure 5 is used to argue that increasing the number of grids increases model performance, but only two grid sizes are used, and it is not clear why the authors choose 7000 grids as the larger size, instead of regular increments increasing from 800 (a size used by Nauck et al., 2022). For these reasons, I vote to reject.
However, these things do not mean the paper is a bad fit for all venues. In fact, I bet this paper would be much better received at an power-generation conference/journal. I’d recommend submitting this work there and then creating a competition for NeurIPS, which has worked out well for past power grid datasets: https://l2rpn.chalearn.org/home